# QTY Code-designed Water-soluble Fc-fusion Cytokine Receptors Bind to their Respective Ligands

**Key words**
Antibody-like fusion protein; cytokine release syndrome; protein design; water-soluble membrane protein

**Author for correspondence:**
Shuguang Zhang, E-mail: Shuguang@mit.edu;
Rui Qing, E-mail: Ruiqing@mit.edu

Shilei Hao[1], David Jin[2], Shuguang Zhang[1] and Rui Qing[1,3]

[1]Media Lab, Massachusetts Institute of Technology, 77 Massachusetts Avenue, Cambridge, MA 02139, USA; [2]Avalon GloboCare Corp., Freehold, New Jersey, USA and [3]The David H. Koch Institute for Integrative Cancer Research, Massachusetts Institute of Technology, 77 Massachusetts Avenue, Cambridge, MA 02139, USA

## Abstract

Cytokine release syndrome (CRS), or 'cytokine storm', is the leading side effect during chimeric antigen receptor (CAR)-T therapy that is potentially life-threatening. It also plays a critical role in viral infections such as Coronavirus Disease 2019 (COVID-19). Therefore, efficient removal of excessive cytokines is essential for treatment. We previously reported a novel protein modification tool called the QTY code, through which hydrophobic amino acids Leu, Ile, Val and Phe are replaced by Gln (Q), Thr (T) and Tyr (Y). Thus, the functional detergent-free equivalents of membrane proteins can be designed. Here, we report the application of the QTY code on six variants of cytokine receptors, including interleukin receptors IL4Rα and IL10Rα, chemokine receptors CCR9 and CXCR2, as well as interferon receptors IFNγR1 and IFNλR1. QTY-variant cytokine receptors exhibit physiological properties similar to those of native receptors without the presence of hydrophobic segments. The receptors were fused to the Fc region of immunoglobulin G (IgG) protein to form an antibody-like structure. These QTY code-designed Fc-fusion receptors were expressed in *Escherichia coli* and purified. The resulting water-soluble fusion receptors bind to their respective ligands with $K_d$ values affinity similar to isolated native receptors. Our cytokine receptor–Fc-fusion proteins potentially serve as an antibody-like decoy to dampen the excessive cytokine levels associated with CRS and COVID-19 infection.

## Introduction

Chimeric antigen receptor (CAR) T-cell therapy is a novel type of cellular immunotherapy in which a patient's T cells are engineered *in vitro* to target and eliminate cancer cells *in vivo*. In CAR-T treatment, the T cells from a patient's blood are extracted by apheresis. The gene for a specific receptor (CAR) which binds to a certain tumor target is delivered to the T cells by viral vector or nonviral transposon methods (Srivastava and Riddell, 2015; Jain and Davila, 2018; Ittershagen *et al.*, 2019). At present, two anti-CD19 CAR-T products have been approved by the U.S. FDA for the treatment of B-cell acute lymphoblastic leukemia and non-Hodgkin lymphoma; CAR-T therapy for other cancer types are undergoing vigorous clinical studies. CAR-T therapy holds great promise for treating hematologic malignancies, and recent clinical evidence has indicated that similar approaches can also be used to treat solid tumors (Baybutt *et al.*, 2019).

However, there are several potentially fatal side effects during CAR-T treatment including: cytokine release syndrome (CRS), neurologic events, neutropenia and anemia (Xu and Tang, 2014). Among all the side effects, CRS is significant and can be life-threatening. Cytokines are essential immune mediators. Yet, a large and rapid release of cytokines into the blood from immune cells can induce a 'cytokine storm', or CRS, which is associated with systemic symptoms of various severity. Most patients with CRS develop a mild flu-like reaction such as fever, fatigue, headache and rash. However, the reaction may progress to an uncontrolled, systemic inflammatory response with extreme pyrexia and become life-threatening (Shimabukuro-Vornhagen *et al.*, 2018).

CRS is not manifested only as a side effect of cellular immunotherapy. It can also be triggered by viral infections such as influenza and hepatitis virus (de Jong *et al.*, 2006; Tisoncik *et al.*, 2012; Savarin and Bergmann, 2018). The current Coronavirus Disease 2019 (COVID-19) triggers CRS in many stages of its pathological course that causes lung fibrosis, acute respiratory distress syndrome, and eventually leads to multi-organ failure (Huang *et al.*, 2020; Xu *et al.*, 2020). Other conditions, including graft-*versus*-host disease, sepsis, Ebola, avian influenza, smallpox and systemic inflammatory response syndrome, also involve extensive release of undesired cytokines (Drazen *et al.*, 2000). To alleviate the symptoms and treat the disease, it is important to remove the excessive cytokines efficiently and rapidly.

The chemokine receptor CCR9, together with its ligand CCL25, contribute to intestinal homing of T cells. This signaling pathway promotes invasion, metastasis, anti-apoptosis and drug-resistance in many types of cancer (Hu *et al.*, 2011; Tu *et al.*, 2016). Specifically, CCR9 is

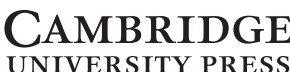

aberrantly expressed in acute and chronic T-cell leukemias that contribute to the aggressiveness of the diseases (Somovilla-Crespo *et al.*, 2018). Targeting CCR9 (such as anti-CCR9 monoclonal antibodies with high specificity, affinity and stability) is a rational therapeutic strategy for these diseases.

We have previously devised a novel tool called 'QTY code' which regulates the water solubility of redesigned membrane proteins through pairwise substitution of hydrophobic amino acids by hydrophilic ones (Zhang *et al.*, 2018). Hydrophobic amino acids Leu, Val, Ile and Phe are exchanged for hydrophilic Gln, Thr and Tyr in the transmembrane regions of a receptor, based on the structural and electron density map similarity in their side chains. It has been shown that this protein design approach enables the solubilization of many types of chemokine receptors with tunable functionality (Qing *et al.*, 2019). The QTY code also provides flexibility in studying the physiological and functional properties of transmembrane receptors as well as promoting their utilization, without the requirements of time consuming and expensive detergent screening or use of nanodisks.

We, here, report the QTY code design of six types of cytokine receptors including two variants of chemokine receptors CCR9 and CXCR2, two variants of interleukin receptors, IL4Rα and IL10Rα, as well as two variants of interferon receptors, IFNγR1 and IFNλR1. These QTY code-designed receptors show ligand-binding properties similar to their native receptor counterparts without the presence of hydrophobic patches. The receptors were fused with Fc domain of mouse IgG2a protein to form an antibody-like structure. These Fc-fusion receptors can be expressed and purified in an *Escherichia coli* system with sufficient yield (~mg/l) in lysogeny broth (LB) media. We also showed that these QTY receptors are capable of binding to their respective ligands with affinity close to isolated native receptors on solution-based assays. These QTY code designs of functional, water-soluble Fc-fusion cytokine receptors can potentially be used clinically as decoy therapy to rapidly remove excessive cytokines in the setting of hyperactive immune reactions during CRS including current COVID-19 severely infected patients.

## Results

### Design of Fc-fusion QTY variant cytokine receptors

Six types of cytokine receptors were selected and redesigned by QTY code. Two variants of chemokine receptors, belonging to the 7-transmembrane (7-TM) G protein-coupled receptor (GPCR) family, and four types single-transmembrane interleukin and interferon receptors were chosen. The L, I, V and F amino acid residues in the transmembrane region of the corresponding receptors were replaced by Q, T and Y accordingly.

Similar to our previous reports for chemokine receptors, sequences of QTY code-designed CCR9 and CXCR2 were aligned with native receptors to compare substitutions of amino acids (Fig. 1*a*,*b*). QTY substitutions were applied to all corresponding residues but only to the 7-TM region. Amino acid changes were highlighted in red color denoting an exchange. Residues in extracellular domain (EC, colored black) or intracellular domain (IC, colored yellow) were untouched. Molecular weight of the QTY variant receptors are slightly increased due to the higher molecular mass of Q, T and Y as compared to L, I, V and F. Despite a total difference of 26.0% (46.4% in 7-TM) in the primary sequence for CCR9$^{QTY}$, and 25.5% (58.9% in 7-TM) for CXCR2$^{QTY}$, the changes in the isoelectric point (pI) of the redesigned proteins were only 0.05 and 0.06 units, respectively. This is attributed to the nonionic

nature of Q, T and Y as these amino acids do not introduce significant changes in the net charge of a protein. Rather, the substituted amino acids form numerous intra-helical and inter-helical hydrogen bonds that contribute to the structural integrity as well as those to the surrounding water molecules that enhance the overall solubility of a protein (Qing *et al.*, 2019).

Both interferon receptors and interleukin receptors have a single-pass transmembrane domain. The ligand binding domain is typically comprised of multiple stranded β-sheets that form two connected anti-parallel β-barrels. The β-barrels are connected to the transmembrane α-helix which is responsible for signal transduction, presumably playing a role in ligand interaction (Richter *et al.*, 2017). In order to best mimic a native receptor, we included the transmembrane domain in our design with a few amino acids in the cytoplasmic region to serve as a short linker so as to optimize the binding and structure of the QTY code modified receptors.

Similar to chemokine receptors, the QTY code was only applied to the transmembrane domains of these receptors, as shown in Fig. 1*c–f*. Amino acid exchanges colored in red are selected to eliminate the hydrophobic patches in the designed receptors. Both extracellular domains and intracellular linkers are untouched. Due to the relative weight of TM region, the changes in molecular weight of interferon and interleukin receptors were minimal. pI changes are 0.00, 0.02, 0.18 and 0.01 for IL4Rα$^{QTY}$, IL10Rα$^{QTY}$, IFNγR1$^{QTY}$ and IFNλR1$^{QTY}$, respectively. The larger pI change in IFNγR1$^{QTY}$ was probably due to its larger deviation towards a charge neutral point as compared to other receptors as shown in Fig. 1 (5.10 for IFNγR1$^{QTY}$ compared to 6.15 for IL4Rα$^{QTY}$, 8.68 for IL10Rα$^{QTY}$ and 8.41 for IFNλR1$^{QTY}$).

We specifically designed the QTY receptor variants to fuse with the Fc region of IgG protein in order to acquire an antibody-like structure. The primary benefit of Fc-fusion is to significantly enhance the half-life of the fused protein in human plasma. It can also improve the safety profile of the fused proteins due to reduced immunogenicity whereas synergistic therapeutic effects from both fusion parts is achievable (Levin *et al.*, 2015). On the other hand, the Fc-fusion is naturally a homodimer through covalent bond. They can be easily tuned to form higher order multimeric states with enhanced stability and efficacy (Czajkowsky *et al.*, 2012). Additionally, Fc-fusion further enhances the solubility of QTY-designed cytokine receptors, and serves as an affinity tag for protein purification, as well as for utilization in affinity-based vehicles (beads) for drug delivery. A spacer was introduced to optimize the conformation of QTY-designed receptors in the heavy chain. We used the Fc region of mouse IgG2a in the specific design as it is the functional equivalent of human IgG1. Mouse IgG is chosen over human IgG due to the consideration of implementing mouse cytokine storm model in subsequent experiments, beyond the scope of the current study. The Fc region can be easily exchanged in future designs. Fig. 2 shows a schematic illustration of these cytokine receptor–Fc complex for the six QTY receptor variants. The structural illustrations of corresponding cytokine receptors were obtained through PDB (Protein Data Bank) where applicable (Thiel *et al.*, 2000; Yoon *et al.*, 2005; Miknis *et al.*, 2010; Moraga *et al.*, 2015; Oswald *et al.*, 2016) or from a homology model (CXCR2) (Kwon, 2010). The molecular weight and pI for each of the cytokine receptor–Fc proteins were also denoted in Fig. 1.

### Bioinformatics analysis

QTY variant protein sequences were analysed using a web-based tool TMHMM Server v2.0 to predict the existence of hydrophobic

(a)

| Name | pI | MW (Kda) | Variation (%) | Variation TM (%) |
|---|---|---|---|---|
| CCR9 | 8.54 | 42.0 | - | - |
| CCR9^QTY | 8.49 | 42.5 | 26.0 | 46.4 |
| CCR9^QTY-Fc | 8.09 | 71.6 | - | - |

```
  1 MTPTDFTSPIPNMADDYGSESTSSMEDYVNFNFTDFYCEKNNVRQFASHFLPPLYWLVFI
    |||||||||||||||||||||||||||||||||||||||||||||||||||..||.||....
    MTPTDFTSPIPNMADDYGSESTSSMEDYVNFNFTDFYCEKNNVRQFASHYQPPQYWQTYT

 61 VGALGNSLVILVYWYCTRVKTMTDMFLLNLAIADLLFLVTLPFWAIAAADQWKFQTFMCK
    .||.|||.||....||||||||||||....|.|.||.....|.|.||.||||||||||||
    TGAQGNSQTTQTYWYCTRVKTMTDMYQQNQATADQQYQTTQPYWATAAADQWKFQTFMCK

121 VVNSMYKMNFYSCVLLIMCISVDRYIAIAQAMRAHTWREKRLLYSKMVCFTIWVLAAALC
    ..|||||||.|||.....||.|.||.||||||AMRAHTWRE|KRQQYSKMTCYTTWTQAAAQC
    TTNSMYKMNYYSCTQQTMCTSTDRYTATAQAMRAHTWREKRQQYSKMTCYTTWTQAAAQC

181 IPEILYSQIKEESGIAICTMVYPSDESTKLKSAVLTLKVILGFFLPFVVMACCYTIIIHT
    .||..|||||||||||||||||||||||KQKSATQTQKTTQGYYQPYTTMACCYTTTTHT
    TPETQYSQIKEESGIAICTMVYPSDESTKQKSATQTQKTTQGYYQPYTTMACCYTTTTHT

241 LIQAKKSSKHKALKVTITVLTVFVLSQFPYNCILLVQTIDAYAMFISNCAVSTNIDICFQ
    ..||||||||.|.|.|..||||||.||||||||||QTQAKKSSKHKAQKTTTTTQTTYTQSQYPYNCTQQTQTTDAYAMFISNCATSTNTDTCYQ
    QTQAKKSSKHKAQKTTTTTQTTYTQSQYPYNCTQQTQTTDAYAMFISNCATSTNTDTCYQ

301 VTQTIAFFHSCLNPVLYVFVGERFRRDLVKTLKNLGCISQAQWVSFTRREGSLKLSSMLL
    .|||.|...|||.||...||||ERFRRDLVKTLKNLGCISQAQWVSFTRREGSLKLSSMLL
    TTQTTAYYHSCQNPTQYTYTGERFRRDLVKTLKNLGCISQAQWVSFTRREGSLKLSSMLL

361 ETTSGALSL
    |||||||||
    ETTSGALSL
```

(b)

| Name | pI | MW (Kda) | Variation (%) | Variation TM (%) |
|---|---|---|---|---|
| CXCR2 | 8.66 | 40.8 | - | - |
| CXCR2^QTY | 8.60 | 41.5 | 25.5 | 58.9 |
| CXCR2^QTY-Fc | 8.20 | 70.6 | - | - |

```
  1 MEDFNMESDSFEDFWKGEDLSNYSYSSTLPPFLLDAAPCEPESLEINKYFVVIIYALVFL
    ||||||||||||||||||||||||||||||||||||||||||||||||||....|....
    MEDFNMESDSFEDFWKGEDLSNYSYSSTLPPFLLDAAPCEPESLEINKYYTTTTYAQTYQ

 61 LSLLGNSLVMLILYSRVGVRSVTDVYLLNLALADLLFALTLPIWAASKVNGWIFGTFLCK
    .|.||||..|....||||||||||||.|.|.|.||.....|.|.|.||||||||||||||
    QSQQGNSQTMQTTQYSRVGVRSVTDTYQQNQAQADQQYAQTQPTWAASKVNGWIFGTFLCK

121 VVSLLKEVNFYSGILLLACISVDRYLAIVHATRTLTQKRYLVKFICLSIWGLSLLIALPV
    ..|.||.|.|||..|.....||DRYLAIVHATRTLTQKRYLVKYTCQSTWGQSQQQAQPT
    TTSQQKETNYYSGTQQQACTSTDRYLAIVHATRTLTQKRYLVKYTCQSTWGQSQQQAQPT

181 LLFRRTVYSSNVSPACYEDMGNNTANWRMLLRILPQSFGFIVPLLIMLFCYGFTLRTLFK
    ...||||||||||||||||||||||||||RMQQRTQPQSYGYTQPQQTMQYCYGFTLRTLFK
    QQYRRTVYSSNVSPACYEDMGNNTANWRMQQRTQPQSYGYTQPQQTMQYCYGFTLRTLFK

241 AHMGQKHRAMRVIFAVVLIFLLCWLPYNLVLLADTLMRTQVIQETCERRNHIDRALDATE
    |||||||||||||||||||||||||||||||.|....|..|.|....|||||||||||AQDATE
    AHMGQKHRAMRRTTYATTQTYQQCWQPYNQTQQADTLMRTQVIQETCERRNHIDRAQDATE

301 ILGILHSCLNPLIYAFIGQKFRHGLLKILAIHGLISKDSLPKDSRPSFVGSSSGHTSTTL
    ..|.|.|||.|....|GQKFRHGLLKILAIHGLISKDSLPKDSRPSFVGSSSGHTSTTL
    TQGTQHSCQNPTQYAYTGQKFRHGLLKILAIHGLISKDSLPKDSRPSFVGSSSGHTSTTL
```

(c)

| Name | pI | MW (Kda) | Variation (%) | Variation TM (%) |
|---|---|---|---|---|
| IL4Rα | 6.15 | 29.4 | - | - |
| IL4Rα^QTY | 6.15 | 29.5 | 5.4 | 60.9 |
| IL4Rα^QTY-Fc | 6.65 | 58.6 | - | - |

```
  1 MGWLCSGLLFPVSCLVLLQVASSGNMKVLQEPTCVSDYMSISTCEWKMNGPTNCSTELRL
    ||||||||||||||||||||||||||||||||||||||||||||||||||||||||||||
    MGWLCSGLLFPVSCLVLLQVASSGNMKVLQEPTCVSDYMSISTCEWKMNGPTNCSTELRL

 61 LYQLVFLLSEAHTCIPENNGGAGCVCHLLMDDVVSADNYTLDLWAGQQLLWKGSFKPSEH
    ||||||||||||||||||||||||||||||||||||||||||||||||||||||||||||
    LYQLVFLLSEAHTCIPENNGGAGCVCHLLMDDVVSADNYTLDLWAGQQLLWKGSFKPSEH

121 VKPRAPGNLTVHTNVSDTLLLTWSNPYPPDNYLYNHLTYAVNIWSENDPADFRIYNVTYL
    ||||||||||||||||||||||||||||||||||||||||||||||||||||||||||||
    VKPRAPGNLTVHTNVSDTLLLTWSNPYPPDNYLYNHLTYAVNIWSENDPADFRIYNVTYL

181 EPSLRIAASTLKSGISYRARVRAWAQCYNTTWSEWSPSTKWHNSYREPFEQHLLLGVSVS
    |||||||||||||||||||||||||||||||||||||||||||||||||||...|.|.|
    EPSLRIAASTLKSGISYRARVRAWAQCYNTTWSEWSPSTKWHNSYREPFEQHQQQGTSTS

241 CIVILAVCLLCYVSITKIKKE
    |....|..|..||.|||||||
    CTTTQATCQQCYTSTTKIKKE
```

(d)

| Name | pI | MW (Kda) | Variation (%) | Variation TM (%) |
|---|---|---|---|---|
| IL10R | 8.70 | 30.0 | - | - |
| IL10Rα^QTY | 8.68 | 30.1 | 4.9 | 61.9 |
| IL10Rα^QTY-Fc | 8.14 | 59.2 | - | - |

```
MLPCLVVLLAALLSLRLGSDAHGTELPSPPSVWFEAEFFHHILHWTPIPNQSESTCYEVA
||||||||||||||||||||||||||||||||||||||||||||||||||||||||||||
MLPCLVVLLAALLSLRLGSDAHGTELPSPPSVWFEAEFFHHILHWTPIPNQSESTCYEVA

LLRYGIESWNSISNCSQTLSYDLTAVTLDLYHSNGYRARVRAVDGSRHSNWTVTNTRFSV
||||||||||||||||||||||||||||||||||||||||||||||||||||||||||||
LLRYGIESWNSISNCSQTLSYDLTAVTLDLYHSNGYRARVRAVDGSRHSNWTVTNTRFSV

DEVTLTVGSVNLEIHNGFILGKIQLPRPKMAPANDTYESIFSHFREYEIAIRKVPGNFTF
||||||||||||||||||||||||||||||||||||||||||||||||||||||||||||
DEVTLTVGSVNLEIHNGFILGKIQLPRPKMAPANDTYESIFSHFREYEIAIRKVPGNFTF

THKKVKHENFSLLTSGEVGEFCVQVKPSVASRSNKGMWSKEECISLTRQYFTVTNVIIFF
||||||||||||||||||||||||||||||||||||||||||||||||||||||.....
THKKVKHENFSLLTSGEVGEFCVQVKPSVASRSNKGMWSKEECISLTRQYFTVTNTTTYY

AFVLLLSGALAYCLALQLYVRRRKK
|.....|...||||.|||||||||||
AYTQQQSGAQAYCQAQQLYVRRRKK
```

(e)

| Name | pI | MW (Kda) | Variation (%) | Variation TM (%) |
|---|---|---|---|---|
| IFNγR1 | 4.92 | 29.9 | - | - |
| IFNγR1^QTY | 5.10 | 30.8 | 5.1 | 73.7 |
| IFNγR1^QTY-Fc | 6.12 | 59.9 | - | - |

```
  1 MALLFLLPLVMQGVSRAEMGTADLGPSSVPTPTNVTIESYNMNPIVYWEYQIMPQVPVFT
    ||||||||||||||||||||||||||||||||||||||||||||||||||||||||||||
    MALLFLLPLVMQGVSRAEMGTADLGPSSVPTPTNVTIESYNMNPIVYWEYQIMPQVPVFT

 61 VEVKNYGVKNSEWIDACINISHHYCNISDHVGDPSNSLWVRVKARVGQKESAYAKSEEFA
    ||||||||||||||||||||||||||||||||||||||||||||||||||||||||||||
    VEVKNYGVKNSEWIDACINISHHYCNISDHVGDPSNSLWVRVKARVGQKESAYAKSEEFA

121 VCRDGKIGPPKLDIRKEEKQIMIDIFHPSVFVNGDEQEVDYDPETTCYIRVYNVYVRMNG
    ||||||||||||||||||||||||||||||||||||||||||||||||||||||||||||
    VCRDGKIGPPKLDIRKEEKQIMIDIFHPSVFVNGDEQEVDYDPETTCYIRVYNVYVRMNG

181 SEIQYKILTQKEDDCDEIQCQLAIPVSSLNSQYCVSAEGVLHVWGVTTEKSKEVCITIFN
    ||||||||||||||||||||||||||||||||||||||||||||||||||||||||||||
    SEIQYKILTQKEDDCDEIQCQLAIPVSSLNSQYCVSAEGVLHVWGVTTEKSKEVCITIFN

241 SSIKGSLWIPVVAALLLFLVLSLVFICFYIKK
    ||||||.|.|..||........|....||||||
    SSIKGSQWTPTTAAQQQYQTQSQTYTCFYIKK
```

(f)

| Name | pI | MW (Kda) | Variation (%) | Variation TM (%) |
|---|---|---|---|---|
| IFNλR1 | 8.42 | 28.6 | - | - |
| IFNλR1^QTY | 8.41 | 28.7 | 5.5 | 66.7 |
| IFNλR1^QTY-Fc | 7.88 | 57.8 | - | - |

```
  1 MAGPERWGPLLLCLLQAAPGRPRLAPPQNVTLLSQNFSVYLTWLPGLGNPQDVTYFVAYQ
    ||||||||||||||||||||||||||||||||||||||||||||||||||||||||||||
    MAGPERWGPLLLCLLQAAPGRPRLAPPQNVTLLSQNFSVYLTWLPGLGNPQDVTYFVAYQ

 61 SSPTRRRWREVEECAGTKELLCSMMCLKKQDLYNKFKGRVRTVSPSSKSPWVESEYLDYL
    ||||||||||||||||||||||||||||||||||||||||||||||||||||||||||||
    SSPTRRRWREVEECAGTKELLCSMMCLKKQDLYNKFKGRVRTVSPSSKSPWVESEYLDYL

121 FEVEPAPPVLVLTQTEEILSANATYQLPPCMPPLDLKYEVAFWKEGAGNKTLFPVTPHGQ
    ||||||||||||||||||||||||||||||||||||||||||||||||||||||||||||
    FEVEPAPPVLVLTQTEEILSANATYQLPPCMPPLDLKYEVAFWKEGAGNKTLFPVTPHGQ

181 PVQITLQPAASEHHCLSARTIYTFSVPKYSKFSKPTCFLLEVPEANWAFLVLPSLLILLL
    ||||||||||||||||||||||||||||||||||||||||||||||||||....||.....
    PVQITLQPAASEHHCLSARTIYTFSVPKYSKFSKPTCFLLEVPEANWAYQTQPSQQTQQQ

241 VIAAGGVIWKTLMGN
    ..||||..||||||||
    TTAAGGTTWKTLMGN
```

**Fig 1.** Protein sequence alignment between natural (Top) and QTY redesigned cytokine receptors. (a) CCR9 *versus* CCR9^QTY; (b) CXCR2 *versus* CXCR2^QTY; (c) IL4Rα *versus* IL4Rα^QTY; (d) IL10Rα *versus* IL10Rα^QTY; (e) IFNγR1 *versus* IFNγR1^QTY and (f) IFNλR1 *versus* IFNλR1^QTY. The substitutions of Q, T, and Y are denoted with '.', while '|' indicates no change in residues between the two sequences. The Q, T and Y amino acid substitutions are colored in red. The N-terminus, extracellular loops are black, transmembrane is blue and intracellular components of the receptors are yellow. Characteristics of native, QTY variant–Fc-fusion receptor proteins' pI, molecular weight, and overall variation rate and that % changes only transmembrane segments are presented.

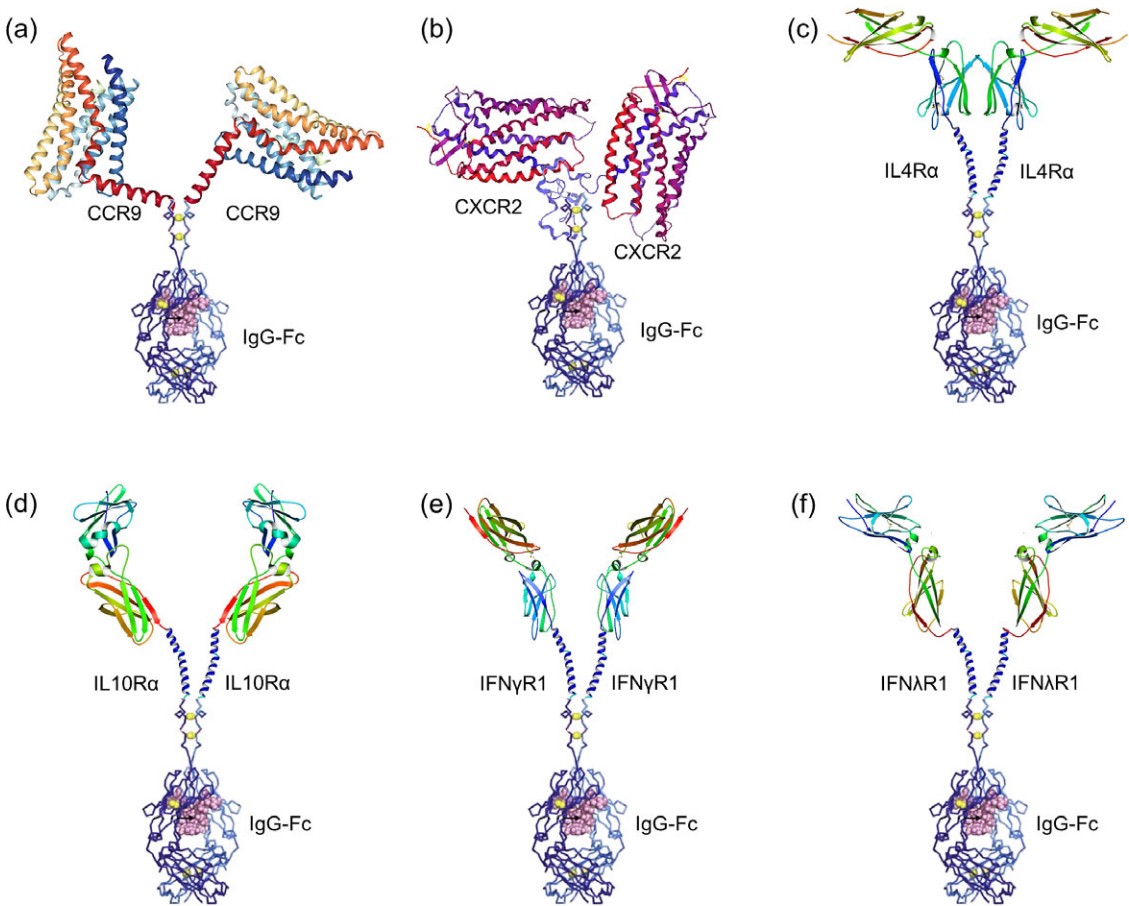

**Fig. 2.** Schematic illustration for Fc-fused QTY variant cytokine receptors with antibody-like structure. (a) CCR9QTY–Fc; (b) CXCR2QTY–Fc; (c) IL4RαQTY–Fc; (d) IL10RαQTY–Fc; (e) IFNγR1QTY–Fc; (f) IFNλR1QTY–Fc. These illustrations are not to scale and the receptors parts are significantly emphasized for clarity.

transmembrane segments. The server is based on a Hidden Markov Model that takes into account actual biological architectures of a transmembrane helix whereas likelihood of presence is calculated (Sonnhammer *et al.*, 1998).

In Fig. 3, the hydrophobicity of a protein is plotted *versus* the protein sequences. The *X*-axis shows the number of amino acids in sequences from N-terminus to C-terminus. Both native sequences (top row) and QTY variant receptor sequences (bottom row) were analysed and aligned. It is apparent that native CCR9 (Fig. 3*a*) and CXCR2 (Fig. 3*b*) exhibits seven distinctive high-probability hydrophobic segments, corresponding to the 7-TM domains. The segments disappear in QTY designed counterparts. In interleukin and interferon receptors, there is only a single high probability hydrophobic segment near the C-terminus end of each receptor which is also eliminated though QTY modification. The hydrophobicity of both extracellular and intracellular components is unchanged.

### E. coli expression and gel-electrophoresis of QTY variant receptors

The corresponding genes with *E. coli* specific codons were synthesized and expressed in sufficient quantities. The throughput for each receptor differed but was all in the mg/l range in LB media. All Fc-fusion receptors were expressed into inclusion bodies. They were purified by (a) affinity purification and (b) gel filtration in denatured state and then folded into functional state for subsequent analysis. Both arginine and dithiothreitol (DTT) were beneficial for

solubilizing the proteins so either or both of them were included in the storage buffer or for ligand binding tests.

The gel-electrophoresis results for purified Fc-fusion QTY variant receptors are shown in Fig. 4. All interleukin and interferon receptors exhibited monomer bands that corresponded well with their respective molecular weight. For the two chemokine receptors, there are several bands above the monomer bands. It is plausible that these bands can be attributed to a dimeric or higher order of multimeric receptors.

### Ligand-binding measurement in buffer

The affinity of QTY modified cytokine receptors fused with Fc of IgG for their respective native ligands was measured using microscale thermophoresis (MST). Changes in thermophoretic movement for labelled proteins upon ligand-binding were recorded and plotted as a function of ligand concentration. Both QTY code-designed Fc-fusion interleukin and interferon receptors showed no nonspecific adhesion or aggregation during the measurement. The binding data were obtained in early T-Jump period, where rapid changes in fluorophore properties induced by fast temperature change was recorded (Jerabek-Willemsen *et al.*, 2014) because two 7-TM chemokine receptors exhibited minor aggregation during prolonged incubation. For better visualization, the data were replotted as bound fraction *versus* concentration with a scale of 0–1, as shown in Fig. 5. The plot was then used to calculate the dissociation constant ($K_d$) value for receptor–ligand interaction using

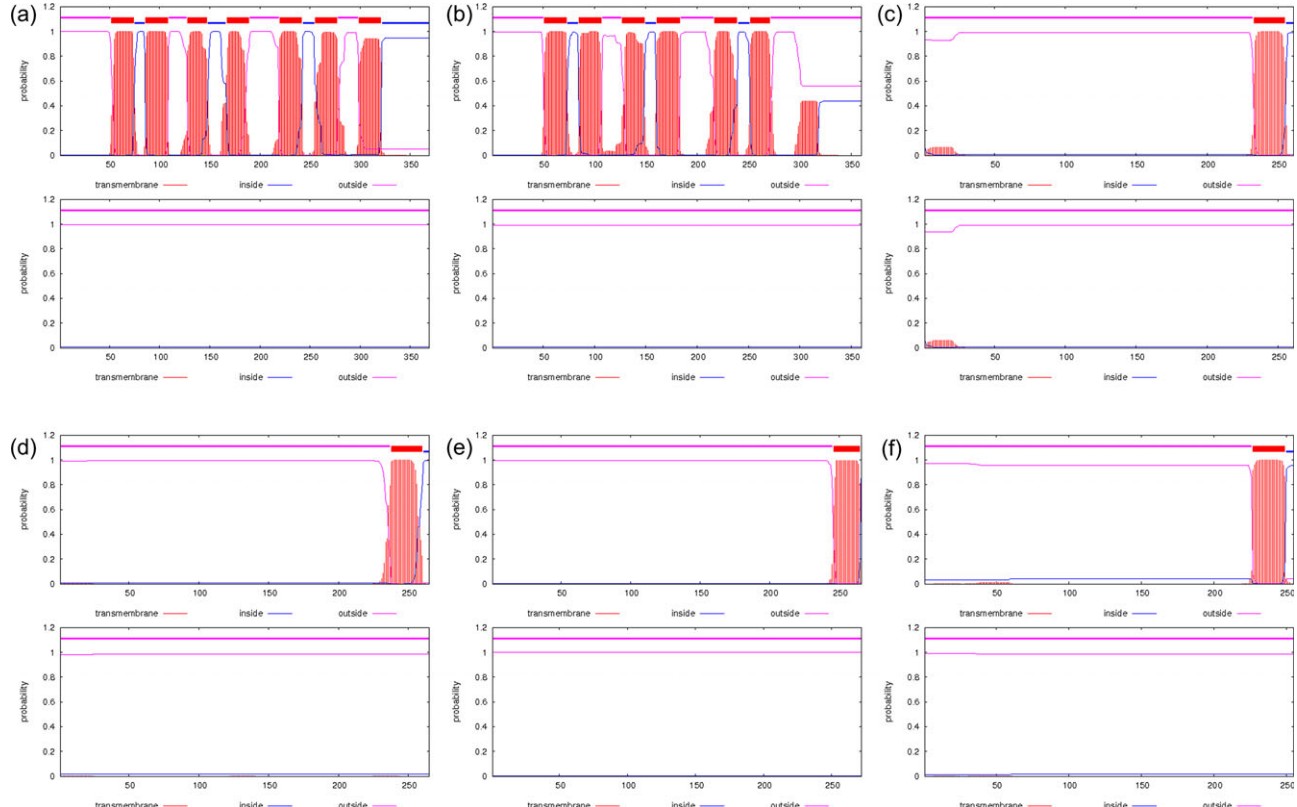

**Fig. 3.** Bioinformatic predictions of cytokine receptors with hydrophobic segment of native (top) and QTY variant (bottom). The hydrophobicity probability of a protein is plotted *vs* the sequence. (a) CCR9$^{QTY}$–Fc; (b) CXCR2$^{QTY}$–Fc; (c) IL4Rα$^{QTY}$–Fc; (d) IL10Rα$^{QTY}$–Fc; (e) IFNγR1$^{QTY}$–Fc and (f) IFNλR1$^{QTY}$–Fc. Color code: pink line, extracellular regions; red line, transmembrane regions and blue line, intracellular region.

the $K_d$ model, as presented in the Materials and Methods section. The fitted curves for $K_d$ calculation are also presented in graphs for illustration.

These QTY Fc-fusion receptors exhibit affinity for their respective ligands typically in a range of up to tens of nM (Table 1). The affinities are lower compared to the native receptors without Fc-fusion. The binding affinity of CXCR2$^{QTY}$–Fc with IL8 (60.3 ± 21.0 nM) is much lower compared to cell-based assay, for example, monomeric IL8 binding with native CXCR2 (0.5 ± 0.3 nM) but closer to IL8 in dimeric state (8.5 ± 2.0 nM) (Rajarathnam *et al.*, 2006). Caccuri *et al.* (2012) also reported a dissociation constant (70 nM) that is similar to our affinity measurement of CXCR2$^{QTY}$–Fc with IL8. For affinities of interleukin and interferon receptors, previously-reported studies primarily used human neutrophil cell-based assays with isotope [125]I-labelled ligand that is significantly more sensitive than using the purified receptors measured by biophysical instrument; thus, they may not be directly comparable. The affinity $K_d$ derived from MST displays similar values compared to previous SPR measurement on purified proteins with the exception of IL4Rα$^{QTY}$–Fc. The method that was used to determine the $K_d$ in literature is also included in Table 1.

Different types of cytokines are aberrantly expressed during CRS in various pathological conditions. Together with many other types of cytokines, levels of the interleukins, IL-8 and IL-10 are elevated over 10 times and 50 times to ~101.7 and ~100.8 pg/ml, respectively, in the peripheral blood in a nonfatal infection of influenza A (H5N1) (de Jong *et al.*, 2006). Infection with *Francisella tularensis* can lead to an accumulation of excessive IFN-γ, IL-10 and IL-8 to respective levels of ~700, ~1 and ~4 ng/ml in the lungs (Sharma

*et al.*, 2011). A more recent study on COVID-19 indicates a high-level expression of IL-6 in the blood with an average of 7 and 12 ng/ml detected in discharged and expired patients, respectively (Ruan *et al.*, 2020). By carefully choosing the vehicle and delivery mechanism such levels of cytokines are surely treatable by our designed Fc-fused QTY cytokine receptors. Further data and information from ongoing research with *in vivo* and *in vitro* cytokine release assays as well as mouse animal model studies will be presented in separate reports.

## Discussion

In this study, the successful design of QTY code-modified variant chemokine receptors CCR9$^{QTY}$ and CXCR2$^{QTY}$ further expands the plausible applicability of such a protein design algorithm on 7-TM GPCRs. Combined with our prior work (Zhang *et al.*, 2018, Qing *et al.*, 2019), our laboratory has successfully designed and engineered eight variants of soluble GPCRs while retaining their physiological and functional properties, including seven variants of chemokine receptors and one variant of olfactory receptor. The general applicability of the QTY code for GPCRs may further promote the study of these previously difficult targets in a functionally equivalent form. It is likely that the QTY code may be generally applied to other types of multi-pass membrane proteins and difficult-to-express proteins. Those studies are ongoing.

Although truncated soluble interleukin and interferon receptors also exist *in vivo*, primarily by cleavage between the extracellular and transmembrane segments, the QTY code is still meaningful for

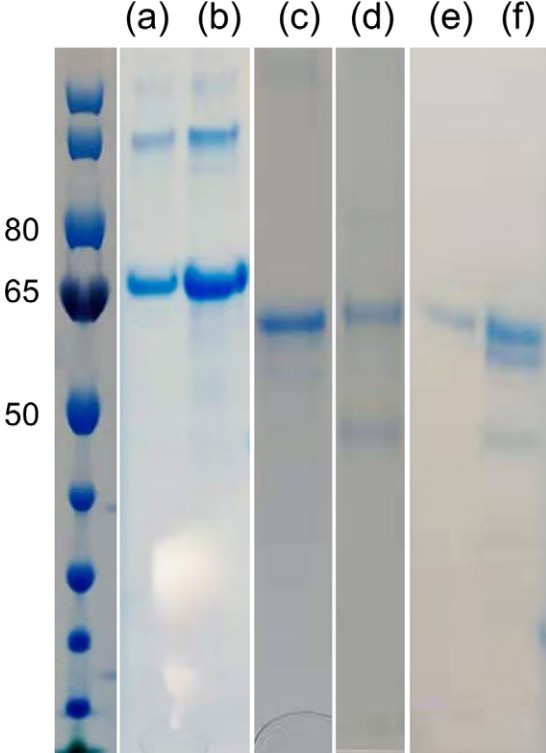

**Fig. 4.** Gel-electrophoresis of purified QTY code designed Fc-fusion receptors. (a) CXCR2$^{QTY}$–Fc; (b) CCR9$^{QTY}$–Fc; (c) IL4Rα$^{QTY}$–Fc; (d) IL10Rα$^{QTY}$–Fc; (e) IFNγR1$^{QTY}$–Fc and (f) IFNλR1$^{QTY}$–Fc. The molecular weight of the ladder is labelled on the left in KDa. It is plausible that these bands can be attributed to dimeric or higher order of multimeric receptors. For panels d and f, the bands are likely to be impurities that are too close to the target band which we were not able to separate with either His-tag or gel-filtration purification. These bands might be eliminated with further Protein A/G purification in future experiments

designing these single transmembrane receptors for additional studies. While for some receptors the transmembrane α-helix appears to play a role in ligand interaction (Richter *et al.*, 2017), inclusion of the QTY-designed water-soluble transmembrane helical segment in the entire receptors may contribute to the overall understanding of receptor functional mechanism and signal transduction. On the other hand, not all native-form soluble interleukin and interferon receptors can be readily synthesized and purified in a high-throughput low cost *E. coli* system. In addition, integral multi-transmembrane cytokine receptors such as CCR9 and CXCR2 do not have correspondingly truncated soluble segments.

Our Fc-fusion water-soluble receptor may be able to rapidly soak up excessive cytokines from a cytokine storm unleashed during CAR-T treatment and COVID-19. When the QTY-designed water-soluble Fc-receptors bind to excessive cytokines, they may inhibit excessive cytokine interaction with target cells, thereby reducing the organ damage and toxicity. There are over 20 Fc-fusion proteins commercially available and several of these have been developed as therapeutics (Czajkowsky *et al.*, 2012). Although there have been many Fc-fusion proteins developed for various applications, they are water-soluble proteins in the native state (Mekhaiel *et al.*, 2011; Czajkowsky *et al.*, 2012). Our QTY code designed Fc-fusion receptors, especially chemokine receptors CCR9 and CXCR2, provide a novel platform for further design of other types of fusion membrane receptors for therapeutic and diagnostic applications.

The QTY code design and synthesis of functionally equivalent transmembrane receptor proteins have implications beyond biological and clinical use. Highly specific membrane receptors towards their respective ligands, QTY code modified transmembrane proteins can also serve as ideal candidates for molecular sensing. Complex electrical arrays functionalized with a variety types of water-soluble membrane proteins can potentially mimic cell response *in vitro* and be fabricated into a pseudo cell with electrical readouts.

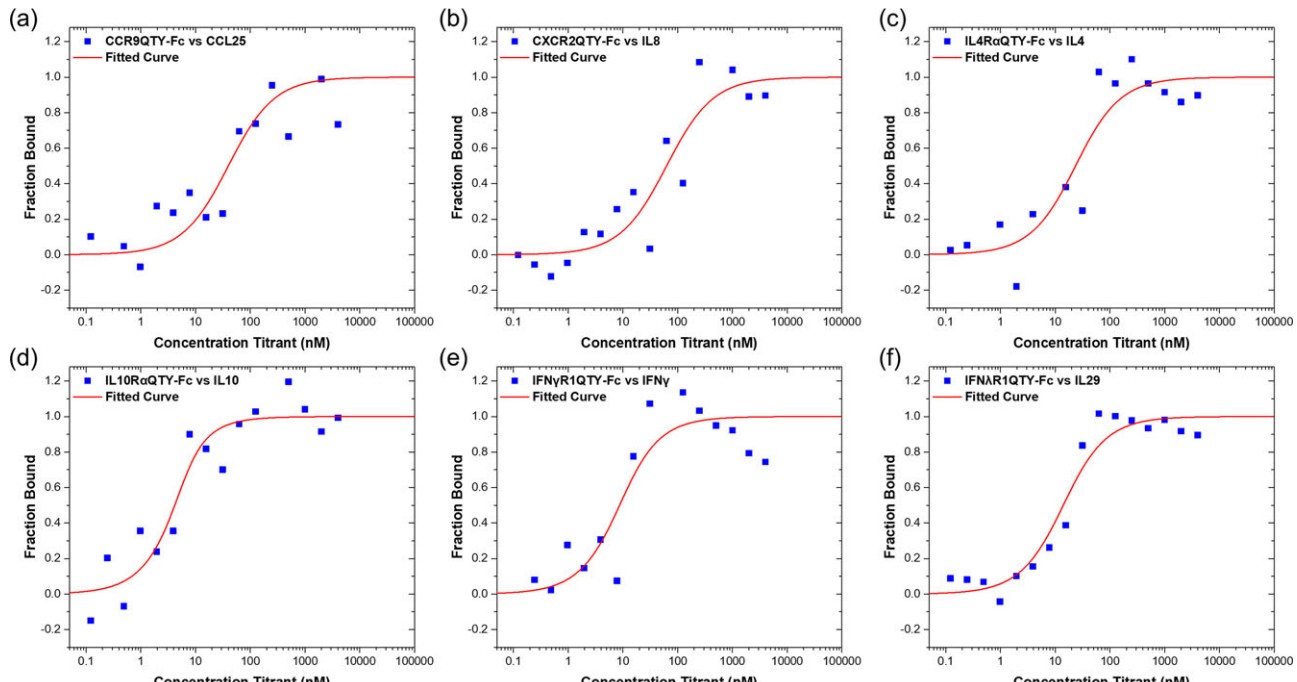

**Fig. 5.** Microscale thermophoresis (MST) ligand-binding measurements. The receptors were labelled with fluorescent dye. The ligands were purchased commercially from and dissolved in deionized water. (a) CCR9$^{QTY}$–Fc with CCL25; (b) CXCR2$^{QTY}$–Fc with IL8; (c) IL4Rα$^{QTY}$–Fc with IL4; (d) IL10Rα$^{QTY}$–Fc with IL10; (e) IFNγR1$^{QTY}$–Fc with IFNγ and (f) IFNλR1$^{QTY}$–Fc with IL29. The $K_d$ values calculated from the graphs are listed in Table 1.

**Table 1.** Ligand-binding affinity of Fc-fused QTY cytokine receptors

| | Native ($K_d$, nM) | QTY variant ($K_d$, nM) |
|---|---|---|
| CCR9$^{QTY}$–Fc *versus* CCL25 | ~8 (Eberhardson *et al.*, 2017) | 37.2 ± 15.7 |
| CXCR2$^{QTY}$–Fc *versus* IL8 | 0.5 ± 0.3 (monomer); 8.5 ± 2.0 (dimer) (Rajarathnam *et al.*, 2006) | 60.3 ± 21.0 |
| IL4R$\alpha^{QTY}$–Fc *versus* IL4 | ~1 nM (SPR) (LaPorte *et al.*, 2008) | 20.9 ± 8.3 |
| IL10R$\alpha^{QTY}$–Fc *versus* IL10 | 0.05–0.2 (cell) (Tan *et al.*, 1993) | 1.6 ± 0.9 |
| IFN$\gamma$R1$^{QTY}$–Fc *versus* IFN$\gamma$ | 1.7 (cell) (Celada *et al.*, 1985); 30.8 (SPR) (Mikulecky *et al.*, 2013) | 6.2 ± 3.0 |
| IFN$\lambda$R1$^{QTY}$–Fc *versus* IL29 | — | 11.5 ± 2.5 |

The methods used for are included after the reported values. The references for reported $K_d$ values are also cited.

## Materials and methods

### Genes identification and QTY modification

Sequences of the selected proteins were obtained from Uniprot: https://www.uniprot.org/. The respective extracellular, transmembrane and cytoplasmic domains were identified. The QTY code was only applied to the transmembrane helical domain to solubilize the proteins.

### Bioinformatics analysis

Protein properties were calculated based on their primary sequences via the open access web-based tool ExPASy: https://web.expasy.org/protparam/. The existence of hydrophobic segments within the transmembrane region in native and QTY variant protein sequences was determined via the open access web-based tool TMHMM Server v.2.0: http://www.cbs.dtu.dk/services/TMHMM-2.0/.

### E. coli expression system and protein purification

Genes of QTY modified cytokine receptor proteins were cloned into the Fc region of mouse IgG2a as the functional equivalent of human IgG1. The full sequences were codon-optimized for *E. coli* expression and obtained from Genscript. The genes were cloned into pET20b expression vector with Carbenicillin resistance. The plasmids were reconstituted and transformed into *E. coli* BL21 (DE3) strain. Transformants were selected on LB medium plates with 100 μg/ml Carbenicillin. *E. coli* cultures were grown at 37°C until the OD$_{600}$ reached 0.4–0.8, after which IPTG (isopropyl-D-thiogalactoside) was added to a final concentration of 1 mM followed by 4-h expression. Cells were lysed by sonication in B-PER protein extraction agent (Thermos-Fisher) and centrifuged (23,000$g$, 40 min, 4°C) to collect the inclusion body. The biomass was then subsequently washed twice in buffer 1 (50 mM Tris-HCl pH 7.4, 50 mM NaCl, 10 mM CaCl2, 0.1%v/v Trition X100, 2 M Urea, 0.2 μm filtered), once in buffer 2 (50 mM Tris-HCl pH 7.4, 1 M NaCl, 10 mM CaCl2, 0.1%v/v Trition X100, 2 M Urea, 0.2 μm filtered) and again in buffer 1. Pellets from each washing step were collected by centrifugation (23,000$g$, 25 min, 4°C).

Washed inclusion bodies were fully solubilized in denaturation buffer [6 M guanidine hydrochloride, 1× phosphate buffered saline (PBS), 10 mM DTT, 0.2 μm filtered] at room temperature for 1.5 h with magnetic stirring. The solution was centrifuged at 23,000$g$ for 40 min at 4°C. The supernatant with proteins was then purified by Qiagen Ni-NTA beads (His-tag) followed by size exclusion chromatography using an ÄKTA Purifier system and a GE healthcare Superdex 200 gel-filtration column. Purified protein was collected and dialysed twice against renaturation buffer (50 mM Tris-HCl pH 9.0, 3 mM reduced glutathione, 1 mM oxidized glutathione, 5 mM ethylenediaminetetraacetic acid and 0.5 M L-arginine). Following an overnight refolding process, the re-natured protein solution was dialysed into storage buffer of 50 mM Tris-HCl pH 9.0 with various arginine content.

### Microscale thermophoresis

MST is an optical method detecting changes in thermophoretic movement and temperature related intensity change of the protein-attached fluorophore upon ligand binding. Active labelled proteins contribute to the thermophoresis signal upon ligand binding. Inactive proteins influence the data as background but not the signals and only data from binding proteins are used to derive the $K_d$ value. Herein ligand binding experiments were carried out with 5 nM NT647-labelled protein in 1× PBS, 10 mM DTT buffer with different concentration of arginine, against a gradient of respective ligands on a Monolith NT.115 pico instrument at 25°C. Synthesized receptors were labelled with Monolith NT second generation protein labelling kit RED – NHS (NanoTemper Technologies) so as to obtain unique fluorescent signals. MST time traces were recorded and analysed to obtain the highest possible signal-to-noise levels and amplitudes, >5 Fnorm units. Multiple rounds of buffer optimization were conducted for CXCR2$^{QTY}$–Fc and CCR9$^{QTY}$–Fc receptors. The data in optimized buffer was reported. The buffer condition was then adopted directly by Fc-fused QTY interleukin and interferon variants. The recorded fluorescence was plotted against the concentration of ligand, and curve fitting was performed using the $K_d$ fit formula derived from the law of mass action. For clarity, binding graphs of each independent experiment were normalized to the fraction bound (0 = unbound, 1 = bound). MST experiments were performed in the Centre for Macromolecular Interactions at Harvard Medical School.

### $K_d$ fitting model

$K_d$ model is the standard fitting model based on law of mass action.
Curve fit formula:

$$F(c_T) = F_u + (F_b - F_u) \times \frac{c_{AT}}{c_A}, \qquad (1)$$

$\frac{c_{AT}}{c_A}$ = fraction bound

$$= \frac{1}{2c_A} \times \left( c_T + c_A + K_d - \sqrt{(c_T + c_A + K_d)^2 - 4c_T c_A} \right), \quad (2)$$

where $F_u$ is the fluorescence in unbound state; $F_b$ the fluorescence in bound state; $K_d$ the dissociation constant, to be determined; $c_{AT}$ the concentration of formed complex; $c_A$ the constant concentration of molecule A (fluorescent), known and $c_T$ the concentration of molecule T in serial dilution.

**Open Peer Review.** To view the open peer review materials for this article, please visit https://doi.org/10.1017/qrd.2020.4.

**Acknowledgements.** This work was primarily funded by Avalon GloboCare Corp. Shilei Hao gratefully acknowledges the fellowship of China Scholarship Council No. 201808505038 and Chongqing University, China. We also thank Dorrie Langsley for her quick turn-around of English editing of this paper.

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
