## [Reviewer Report]

*Comments to Author*: This manuscript is a very interesting follow-up on previous reports from Zhang and coworkers on the QTY code and its application to solubilize membrane protein receptors. Here, QTY variants of cytokine receptors were further fused to the Fc domain of (a mouse) IgG. The authors highlight the potential use of these fusion constructs for treatment of various cytokine disorders, which is of great medical relevance. In essence, the authors suggest that the receptor-Fc fusion proteins could serve as antibody-like decoys to reduce cytokine levels. Naturally, one would like to know if this really works, but in the spirit of the mission of QRB Discovery, I believe that this research merits publication at the present stage provided that the underlying data are sufficiently convincing. In this regard, the present version the manuscript is somewhat lacking.

The most central result in this manuscript is the binding assay aimed to prove that the Fc-fusion QTY variants are functional and bind their targets. Thus, it is critical that these experiments are reproducible and provide accurate estimates of the binding affinities. Compared to previous results reported in Fig. 3 of Zhang et al. (2018) PNAS 115, E8652, the MST data in the present paper (Fig. 5) are of much lower quality. Please describe how many replicates were performed to generate the data in Fig. 5 and please add error bars to the individual data points. Describe how you estimated the uncertainties of the Kd values reported in Table 1.

Further, please report whether negative control experiments were performed, and if so, what type of protein was used as control.

Please describe the rationale behind the receptor-Fc fusion design. At present, you merely state that this was done to “form an antibody-like structure”. Are there any immunological considerations behind this choice? Do you expect the Fc region to bind to the corresponding Fc receptor? (What purpose would this serve?)

It would strengthen the manuscript if you could provide calculations to back up the expectations that the receptor-Fc fusion proteins might be used in the clinic to effectively reduce aberrantly high cytokine levels. How high levels of cytokines are present during a “cytokine storm”? Is it feasible to administer sufficiently high levels of the fusion proteins to the patient in order for the treatment to have a significant effect?

Minor points:

p. 4, 3rd paragraph: I cannot see how both of the following 2 statements can be true: “QTY code was only applied to the transmembrane domains” and (two lines down) “Both extracellular domains (colored black) and intracellular linkers (colored yellow) are redesigned according to the QTY code algorithm”. Am I missing something?

Fig. 4: There are extra bands in panels D and F as well as in A and B. Please comment on these.

Fig. 5 is not mentioned in the paragraph that presents these data (p. 5).

---

## [Reviewer Report]

*Comments to Author*: The manuscript by Shilei Hao is highly interesting and could be a valuable step forward in treating a severe medical problem. The successful development of water-soluble receptors for an excess of cytokines and interferons to manage cytokine storms is an interesting approach. If I am correctly informed humans already have water-soluble cytokine receptors. I find no mention of these in the manuscript. The manuscript describes the work well and the results show a significant success. The explanation for attaching the receptors to Fc parts of immunoglobulins would benefit from being clearer. The manuscript gives no indication for how the soluble receptors can be used to treat the severe problems at cytokine storm. This would be of interest for many readers.

A technical editor could deal with minor linguistic problems. I found a number of “the” and “of” missing.

---

## [Reviewer Report]

*Comments to Editor*: Dear Lynet, We should invite the authors to revise their ms (minor revision). In view of the linguistic glitches that Liljas indicated it would be good if a copy editor could keep an eye open for typos and such things.

Best wishes

Bengt

*Comments to Author*: Reviewer #1: The manuscript by Shilei Hao is highly interesting and could be a valuable step forward in treating a severe medical problem. The successful development of water-soluble receptors for an excess of cytokines and interferons to manage cytokine storms is an interesting approach. If I am correctly informed humans already have water-soluble cytokine receptors. I find no mention of these in the manuscript. The manuscript describes the work well and the results show a significant success. The explanation for attaching the receptors to Fc parts of immunoglobulins would benefit from being clearer. The manuscript gives no indication for how the soluble receptors can be used to treat the severe problems at cytokine storm. This would be of interest for many readers.

A technical editor could deal with minor linguistic problems. I found a number of “the” and “of” missing.

Reviewer #2: This manuscript is a very interesting follow-up on previous reports from Zhang and coworkers on the QTY code and its application to solubilize membrane protein receptors. Here, QTY variants of cytokine receptors were further fused to the Fc domain of (a mouse) IgG. The authors highlight the potential use of these fusion constructs for treatment of various cytokine disorders, which is of great medical relevance. In essence, the authors suggest that the receptor-Fc fusion proteins could serve as antibody-like decoys to reduce cytokine levels. Naturally, one would like to know if this really works, but in the spirit of the mission of QRB Discovery, I believe that this research merits publication at the present stage provided that the underlying data are sufficiently convincing. In this regard, the present version the manuscript is somewhat lacking.

The most central result in this manuscript is the binding assay aimed to prove that the Fc-fusion QTY variants are functional and bind their targets. Thus, it is critical that these experiments are reproducible and provide accurate estimates of the binding affinities. Compared to previous results reported in Fig. 3 of Zhang et al. (2018) PNAS 115, E8652, the MST data in the present paper (Fig. 5) are of much lower quality. Please describe how many replicates were performed to generate the data in Fig. 5 and please add error bars to the individual data points. Describe how you estimated the uncertainties of the Kd values reported in Table 1.

Further, please report whether negative control experiments were performed, and if so, what type of protein was used as control.

Please describe the rationale behind the receptor-Fc fusion design. At present, you merely state that this was done to “form an antibody-like structure”. Are there any immunological considerations behind this choice? Do you expect the Fc region to bind to the corresponding Fc receptor? (What purpose would this serve?)

It would strengthen the manuscript if you could provide calculations to back up the expectations that the receptor-Fc fusion proteins might be used in the clinic to effectively reduce aberrantly high cytokine levels. How high levels of cytokines are present during a “cytokine storm”? Is it feasible to administer sufficiently high levels of the fusion proteins to the patient in order for the treatment to have a significant effect?

Minor points:

p. 4, 3rd paragraph: I cannot see how both of the following 2 statements can be true: “QTY code was only applied to the transmembrane domains” and (two lines down) “Both extracellular domains (colored black) and intracellular linkers (colored yellow) are redesigned according to the QTY code algorithm”. Am I missing something?

Fig. 4: There are extra bands in panels D and F as well as in A and B. Please comment on these.

Fig. 5 is not mentioned in the paragraph that presents these data (p. 5).